# Simple and Scalable Sparse *k*-means Clustering via Feature Ranking

**Zhiyue Zhang**[1]  **Kenneth Lange**[2]  **Jason Xu**[1,‡]
[1]Department of Statistical Science, Duke University
[2] Departments of Computational Medicine, Statistics, and Human Genetics, UCLA
‡Correspondence to `jason.q.xu@duke.edu`

## Abstract

Clustering, a fundamental activity in unsupervised learning, is notoriously difficult when the feature space is high-dimensional. Fortunately, in many realistic scenarios, only a handful of features may be relevant in distinguishing clusters. This has motivated the development of sparse clustering techniques that typically rely on *k*-means within outer algorithms of high computational complexity. Current techniques also require careful tuning of shrinkage parameters, further limiting their scalability. In this paper, we propose a novel framework for sparse *k*-means clustering that is intuitive, simple to implement, and competitive with state-of-the-art algorithms. We show that our algorithm enjoys consistency and convergence guarantees. Our core method readily generalizes to several task-specific algorithms such as clustering on subsets of attributes and in partially observed data settings. We showcase these contributions thoroughly via simulated experiments and real data benchmarks, including a case study on protein expression in trisomic mice.

## 1  Introduction

Clustering is a ubiquitous task in unsupervised learning and exploratory data analysis. First proposed by Steinhaus [1], *k*-means clustering is one of the most widely used approaches owing to its conceptual simplicity, speed, and familiarity. The method aims to partition $n$ data points into $k$ clusters, with each data point assigned to the cluster with nearest center (mean). More recently, variants and extensions of *k*-means continue to be developed from a number of perspectives including Bayesian and nonparametric approaches [2–4], subspace clustering [5, 6], and continuous optimization [7–9]. After decades of refinements and generalizations, Lloyd's algorithm [10] remains the most popular algorithm to perform *k*-means clustering.

**Lloyd's algorithm**  Consider a dataset $\boldsymbol{X} \in \mathbb{R}^{n \times p}$ comprised of $n$ samples and $p$ features. Without loss of generality, assume each feature is centered at the origin. To put features on the same footing, it is also recommended to standardize each of them to have unit variance [11]. Given a fixed number of $k$ clusters, *k*-means assigns each row $\boldsymbol{x}_i^t$ of $\boldsymbol{X}$ to a cluster label $C_j$ represented by optimal cluster centers $\boldsymbol{\theta}_j$ that minimize the objective

$$\sum_{j=1}^{k} \sum_{\boldsymbol{x}_i \in C_j} \|\boldsymbol{x}_i - \boldsymbol{\theta}_j\|_2^2 = \sum_{i=1}^{n} \min_{1 \le j \le k} \|\boldsymbol{x}_i - \boldsymbol{\theta}_j\|_2^2. \tag{1}$$

Although the objective function is simple and intuitive, finding its optimal value is NP-hard, even when $k = 2$ or $p = 2$ [12–14]. A greedy algorithm, Lloyd's *k*-means algorithm alternates between two steps until cluster assignments stabilize and no further changes occur in the objective function: 1) The algorithm assigns each row $\boldsymbol{x}_i^t$ of $\boldsymbol{X}$ to the cluster $C_j$ whose center $\boldsymbol{\theta}_j$ is closest. 2) It then redefines each center $\boldsymbol{\theta}_j$ as the center of mass $\boldsymbol{\theta}_j = \frac{1}{|C_j|} \sum_{i \in C_j} \boldsymbol{x}_i$. While guaranteed to converge in finite iterations, Lloyd's algorithm may stop short at a poor local minimum, rendering it extremely

sensitive to the initial starting point. Many initialization methods have been proposed to ameliorate this defect [15–17]; the most popular scheme to date is called $k$-means++ [18, 19].

**Feature selection and sparsity**    Despite its efficacy and widespread use, Lloyd's $k$-means algorithm is known to deteriorate when the number of features $p$ grows. In the presence of many uninformative features, the signal-to-noise ratio decreases, and the algorithm falls victim to poor local minima. In essence, the pairwise Euclidean distances appearing in the $k$-means objective (1) become less informative as $p$ grows [20, 21]. One way of mitigating this curse of dimensionality is to cluster under a sparsity assumption. If only a subset of features distinguishes between clusters, then ignoring irrelevant features will not only alleviate computational and memory demands, but will also improve clustering accuracy. Feature selection may be desirable as an interpretable end in itself, shedding light on the underlying structure of the data.

Feature selection for clustering is an active area of research, with most of the well-known methods falling under the category of filter methods or wrapper models [22, 23]. Filter methods independently evaluate the clustering quality of the features without applying any clustering algorithm. Feature evaluation can be univariate [24] or multivariate [25]. Filter criteria include entropy-based distances [26] and Laplacian scores [27]. Wrapper model methods (also called hybrid methods) first select subsets of attributes with or without filtering, then evaluate clustering quality based on the candidate subsets, and finally select the subset giving to the highest clustering quality. This process can be iterated until a desired clustering quality is met. The wrapper method of Dy and Brodley [28] uses maximum likelihood criteria and mixture of Gaussians as the base clustering method. A variation of $k$-means called Clustering on Subsets of Attributes (COSA) [29] performs feature weighing to allow different clusters to be determined based by different sets of features. As formulated, COSA does not lead to sparse solutions.

Expanding on COSA, Witten and Tibshirani [30] proposed a framework for feature selection that can result in feature-sparse centers. Their sparse $k$-means method (SKM) aims to maximize the objective

$$\sum_{m=1}^{p} w_m \left( \frac{1}{n} \sum_{i=1}^{n} \sum_{j=1}^{n} d_{ijm} - \sum_{l=1}^{k} \frac{1}{|C_l|} \sum_{i,j \in C_l} d_{ijm} \right) \qquad (2)$$

with respect to the clusters $C_l$ and the feature weights $w_m$ subject to the constraints $\|\boldsymbol{w}\|_2 \leq 1$, $\|\boldsymbol{w}\|_1 \leq s$, and $w_m \geq 0$ for all $m$. Here $d_{ijm}$ denotes a dissimilarity measure between $x_{im}$ and $x_{jm}$, the most natural choice being $d_{ijm} = (x_{im} - x_{jm})^2$. The $\ell_1$ penalty $\|\boldsymbol{w}\|_1 \leq s$ on the weight vector $\boldsymbol{w}$ ensures a sparse selection of the features, while the $\ell_2$ penalty $\|\boldsymbol{w}\|_2 \leq 1$ on $\boldsymbol{w}$ gives more smoothness to the solution. Without smoothness, often only one component of $\boldsymbol{w}$ is estimated as positive. Constrained maximization of the objective (2) is accomplished by block descent: holding $\boldsymbol{w}$ fixed, the objective is optimized with respect to the clusters $C_l$ by the $k$-means tactic of assigning $\boldsymbol{x}_i$ to the cluster with the closest center. In this context the weighted distance $\|\boldsymbol{y}\| = \sum_m w_m y_m^2$ is operative. Holding the $C_l$ fixed, the objective is then optimized with respect to $\boldsymbol{w}$. To handle outliers, Kondo et al. [31] propose trimming observations, and Brodinová et al. [32] assigns weights to observations. Such approaches with convex norm penalties or constraints on feature weights are popular since the work of Huang et al. [33], and remain active areas of research [34, 35].

**Proposed contributions**    We propose a novel method that incorporates feature selection within Lloyd's algorithm by adding a sorting step. This simple adjustment preserves the algorithm's speed and scalability while addressing sparsity directly. Called Sparse $k$-Means with Feature Ranking (SKFR), the method forces all cluster centers $\boldsymbol{\theta}_j$ to fall in the $s$-sparse ball $\mathcal{B}_0(s) := \{\boldsymbol{x} \in \mathbb{R}^p, \|\boldsymbol{x}\|_0 \leq s\}$, with nonzero entries occurring along a shared set of feature indices. One may interpret the algorithm as switching between learning the most informative set of features in $\mathcal{B}_0(s)$ via ranking the components of $\boldsymbol{\theta}_j$ according to their reduction of the loss, and then projecting $\boldsymbol{\theta}_j$ onto this shared sparsity set at each iteration. Computing the projection is simple, and amounts to truncating all but the top $s$ ranked components to $0$. Importantly—and perhaps surprisingly—we will show that this ranking criterion can be computed *without* making a second pass through the data.

This feature ranking and truncation yields *interpretable* sparsity in $\boldsymbol{\theta}_j$, in contrast to $\ell_1$ penalty approaches that promote sparsity by proxy. The latter are known to introduce unwanted shrinkage, and may not always yield sparse solutions. The role of "interpretability" here is twofold: first at the input stage, the parameter $s$ directly informs and controls the sparsity level, and is thus directly

interpretable in contrast to tuning parameters in shrinkage approaches. Second, SKFR selects relevant features among the original dimensions, thus allowing them to retain their original interpretations in that feature space . For instance, our case study on mouse protein expression not only clusters the data but identifies the *most relevant genes* in expression space for predicting trisomy. In this sense the *dimension reduction* is interpretable, in contrast to generic dimension reduction such as PCA, where the axes (principal components) in the projected space no longer can be interpreted as, say, genes. The simplicity of the SKFR framework leads to straightforward modifications that extend to many salient data settings, allowing the method to accommodate for missing values, outliers, and cluster specific sparsity sets.

The rest of the paper is organized as follows. The proposed algorithms are detailed in Section 2, and its properties are discussed in Section 2.1. We demonstrate the flexibility of our core routine by deriving various extensions in Section 3. The methods are validated empirically via a suite of simulation studies and real data in Sections 4 and 5, followed by discussion.

## 2 Sparse *k*-means with feature ranking

We present two versions of the SKFR algorithm, which both begin by standardizing each feature to have zero mean and unit variance. The first version, which we will refer to as SKFR1, chooses an optimal set of $s$ globally informative features based on a ranking criterion defined below. Each informative component $\theta_{jl}$ of cluster $j$ is then set equal to the within-cluster mean $\mu_{jl} = \frac{1}{|C_j|} \sum_{i \in C_j} x_{il}$, and the remaining $p - s$ uninformative features are left at zero.

A second *local* version SKFR2 will allow for the set of relevant features to vary across clusters: the ranking of top $s$ features is performed for each cluster—akin to the COSA algorithm [29]—to permit richer distinctions between clusters. Within each cluster $j$, the $s$ informative components are again updated by setting $\theta_{jl}$ equal to $\mu_{jl}$ with remaining uninformative components left at zero. Both versions of SKFR alternate cluster reassignment with cluster re-centering. In cluster reassignment, each sample is assigned to the cluster with closest center, exactly as in Lloyd's algorithm .

The choice of top features are determined based on the amount they affect the *k*-means objective. For SKFR1 (the global version), the difference

$$d_l = \sum_{i=1}^{n} (x_{il} - 0)^2 - \sum_{j=1}^{k} \sum_{i \in C_j} (x_{il} - \mu_{jl})^2 = -\sum_{j=1}^{k} \mu_{jl}^2 |C_j| + 2 \sum_{j=1}^{k} \mu_{jl} \sum_{i \in C_j} x_{il} = \sum_j |C_j| \mu_{jl}^2 \quad (3)$$

measures the reduction of the objective as feature $l$ passes from uninformative to informative status. The final equality in (3) shows that $d_l$ can be written in a form that does not involve $x_{il}$, which suggests that it can be computed only in terms of the current means and their relative sizes. In particular, it suggests we do *not* require taking a second pass through the data to perform the ranking step. The $s$ largest differences $d_l$ identify the informative features. Alternatively, one can view the selection process as a projection to a sparsity set as mentioned previously. This simple observation provides the foundation to the transparent and efficient Algorithm 1.

In the local version SKFR2, the choice of informative features for cluster $j$ depends on the differences

$$d_{jl} = \sum_{i \in C_j} (x_{il} - 0)^2 - \sum_{i \in C_j} (x_{il} - \mu_{jl})^2 = |C_j| \mu_{jl}^2. \quad (4)$$

The $s$ largest differences $d_{jl}$ identify the informative features for cluster $j$. These informative features may or may not be shared with other clusters. The pseudocode summary appears as Algorithm 2.

As we detail in the next section, both versions of SKFR enjoy a descent property and are guaranteed to reduce the *k*-means objective at each iteration. Both versions dynamically choose informative features, while retaining the greedy nature and low $O(npk)$ computational complexity of Lloyd's algorithm. Like Lloyd's algorithm, neither version is guaranteed to converge to a global minimum of the objective, though they are both guaranteed to converge in finite iterations.

We argue our framework is more straightforward and interpretable than existing alternatives. For instance, the SKM algorithm entails solving constrained subproblems with an inner bisection algorithm at each step to select dual parameters, and encourages sparsity through a less interpretable tuning constant $\lambda$ [30]. The extensions for robust versions of such methods, for instance trimmed

versions, entail even more computational overhead. In contrast, as we see in Section 3, robustification and other modifications accounting for specific data settings will only require natural and simple modifications of the core SKFR algorithm.

---

| **Algorithm 1** SKFR1 algorithm pseudocode | **Algorithm 2** SKFR2 algorithm pseudocode |
|---|---|
| **Input:** data $\boldsymbol{X} \in \mathbf{R}^{n \times p}$, number of clusters $k$, sparsity level $s$, initial clusters $C_j$ | **Input:** data $\boldsymbol{X} \in \mathbf{R}^{n \times p}$, number of clusters $k$, sparsity level $s$, initial clusters $C_j$ |
| **repeat** | **repeat** |
|   **for** each cluster $j$: **do** |   **for** each cluster $j$: **do** |
|     $\boldsymbol{\mu}_j = \frac{1}{\|C_j\|} \sum_{x_i \in C_j} \boldsymbol{x}_i$ |     $\boldsymbol{\mu}_j = \frac{1}{\|C_j\|} \sum_{x_i \in C_j} \boldsymbol{x}_i$ |
|   **end for** |     **for** each feature $l$: **do** |
|   **for** each feature $l$: **do** |       Rank $l$ by criterion $d_{jl} = \|C_j\|\mu_{jl}^2$ |
|     Rank $l$ by criterion $d_l = \sum_j \|C_j\|\mu_{jl}^2$ |     **end for** |
|   **end for** |     Let $L_j$ be the set of features $l$ with rank($l$) $\leq s$ |
|   Let $L$ be the set of features $l$ with rank($l$) $\leq s$ |   **end for** |
|   **for** each sample $i$: **do** |   **for** each sample $i$: **do** |
|     Assign $\boldsymbol{x}_i$ to the cluster $C_j$ such that $j$ minimizes $\sum_{l \in L}(x_{il} - \mu_{jl})^2 + \sum_{l \notin L} x_{il}^2$ |     Assign $\boldsymbol{x}_i$ to cluster $C_j$ such that $j$ minimizes $\sum_{l \in L_j}(x_{il} - \mu_{jl})^2 + \sum_{l \notin L_j} x_{il}^2$. |
|   **end for** |   **end for** |
| **until** convergence | **until** convergence |

## 2.1 Properties

Here we discuss several theoretical properties of the algorithm. Complete proofs are detailed in the Supplement. We first observe that in the extreme case when $s = p$, our algorithm reduces to Lloyd's original $k$-means algorithm. As the sorting step in ranking is only logarithmic in cost, it is straightforward to see that our algorithm enjoys the same $O(npk)$ complexity overall.

**Monotonicity of the objective**    To establish convergence of the algorithm, it suffices to prove that it possesses a descent property as the objective is bounded below. Monotonicity also lends itself to stability of the algorithm in practice.

**Proposition 1.** *Each iteration of Algorithm 1 and Algorithm 2 monotonically decreases the $k$-means objective function $h(C, \boldsymbol{\theta}) = \sum_{j=1}^{k} \sum_{\boldsymbol{x} \in C_j} \|\boldsymbol{x} - \boldsymbol{\theta}_j\|_2^2$.*

**Strong consistency of centroids**    We additionally establish convergence in the statistical sense in that our method enjoys strong consistency in terms of the optimal centroids. In particular, Proposition 2 shows that our method inherits a property of standard $k$-means method established by Pollard [36]. Let $\{\boldsymbol{x}\}_n$ be independently sampled from a distribution $P$, and let $\Theta \subset \mathcal{B}_0(s)$ denote a set of $k$ points. Define the population-level loss function

$$\Phi(\Theta, P) = \int \min_{\boldsymbol{\theta} \in \Theta} \|\boldsymbol{x} - \boldsymbol{\theta}\|^2 P(d\boldsymbol{x}). \tag{5}$$

Let $\Theta^*$ denote the minimizer of $\Phi(\Theta, P)$, and let $\Theta_n$ be the minimizer of $\Phi(\Theta, P_n)$, where $P_n$ is the empirical measure. The following proposition establishes strong consistency under an identifiability assumption and a selection consistency assumption.

**Proposition 2.** *Assume that for any neighborhood $\mathcal{N}$ of $\Theta^*$, there exists $\eta > 0$ such that $\Phi(\Theta, P) > \Phi(\Theta^*, P) + \eta$ for every $\Theta \notin \mathcal{N}$. Then if $\Theta_n$ eventually lie in the same dimension as $\Theta^*$ (i.e., the nonzero feature indices of $\Theta_n$ agree with $\Theta^*$ as $n \to \infty$), we have $\Theta_n \xrightarrow{a.s.} \Theta^*$ as $n \to \infty$.*

We remark that while the identifiability of $\Theta^*$ is a mild condition, consistency of cluster centers relies on the top $s$ features to be identified correctly. The latter is a considerably strong assumption, and it will be of interest to investigate conditions in the data that allow us to relax the assumption.

# 3 Extensions to SKFR

The simplicity of the SKFR framework allows for straightforward, modular extensions of our two basic algorithms. In the scenarios considered below, existing variations on $k$-means often require complex adjustments that must be derived on a case-by-case basis. Our simple core routine applies to a variety of common situations in data analysis.

**Missing data**    Missing data are common in many real-world data applications. To recover a complete dataset amenable to traditional clustering, practitioners typically impute missing features or delete incomplete samples in a preprocessing step. Imputation can be computationally expensive and potentially erroneous. Deleting samples with missing features risks losing valuable information and is ill advised when the proportion of samples with missing data is high. Chi et al. [7] proposed a simple alternative called $k$-pod that does not rely on additional tuning parameters or assumptions on the pattern of missingness. Let $\Omega \subset \{1,...n\} \times \{1,...,p\}$ denote the index set corresponding to the observed entries of the data matrix $\boldsymbol{X}$. The $k$-means objective is rephrased here as $\sum_{j=1}^{k} \sum_{i \in C_j} \sum_{(i,l) \in \Omega} (x_{il} - \theta_{jl})^2$.

Their strategy invokes the majorization-minimization (MM) principle [37], which involves majorizing the objective by a surrogate function at each iteration and then minimizing the surrogate. This action provably drives the objective downhill [38]. At iteration $n$ define $y_{n,il}$ by $x_{il}$ for $(i,l) \in \Omega$ and by $\theta_{n,jl}$ for $(i,l) \notin \Omega$ and $\boldsymbol{x}_i$ assigned to cluster $j$. The surrogate function is then defined by $\sum_{i=1}^{n} \min_{1 \le j \le k} \|\boldsymbol{y}_i - \boldsymbol{\theta}_j\|_2^2$. In other words, each missing $x_{il}$ is filled in by the current best guess of its value. Because our sparse clustering framework shares the same skeleton as $k$-means, exactly the same imputation technique carries over. Every iteration under imputation then makes progress in reducing the objective, preserving convergence guarantees.

**Outliers**    One popular way to deal with outlying data is via data *trimming*, but due to its additional computational overhead, this approach becomes quickly limited to only moderate dimensional settings [31, 32]. A viable alternative to handle outliers is to replace squared Euclidean norms by a robust alternative. The $\ell_1$ norm is such a choice, which entails replacing the within-cluster means of SKFR1 by within-cluster medians, and the global distances (3) by

$$d_l = \sum_{i=1}^{n} |x_{il} - 0| - \sum_{j=1}^{k} \sum_{i \in C_j} |x_{il} - \mu_{jl}|.$$

An analogous expression holds for the local version of SKFR. Observe that the revised distances lose some of the algebraic simplicity of the original distances; now we must pass through the data to compute $d_l$. Additionally, the user may choose not to standardize the data with respect to squared distance beforehand to further reduce outlier effects [39]. The overall algorithm remains identical to SKFR up to this substitution for $d_l$.

**Choice of sparsity level**    In contrast to penalized methods, SKFR deals with sparsity directly through an interpretable sparsity level $s$. If one desires a particular level $s$ or $s$ is known in advance, then it does not require tuning. In sparse $k$-means (SKM) for instance [30], prior information does not directly inform the choice of the continuous shrinkage $\lambda$. When the sparsity level $s$ must be learned, one can capitalize on a variant of the gap statistic [40] to find an optimal value of $s$. Witten and Tibshirani [30] suggest a permutation test and a score closely related to the original gap statistic to tune the $\ell_1$ bound parameter on the feature weight vector. Here we adopt a similar approach, employing the gap statistic which depends on the difference

$$O(\boldsymbol{X}, s) = \sum_{\boldsymbol{x}_i \in \boldsymbol{X}} \|\boldsymbol{x}_i - \bar{\boldsymbol{x}}\|_2^2 - \sum_{j=1}^{k} \sum_{i \in C_j} \|\boldsymbol{x}_i - \boldsymbol{\theta}_j\|_2^2$$

between the total sum of squares and the sums of the within-cluster sum of squares at the optimal cluster. One now randomly permutes the observations within each column of $\boldsymbol{X}$ to obtain $B$ independent datasets $\boldsymbol{X}_1, \ldots, \boldsymbol{X}_B$. Finally, the parameter $s$ is chosen to maximize

$$\text{Gap}(s) = \log O(\boldsymbol{X}, s) - \frac{1}{B} \sum_{b=1}^{B} \log O(\boldsymbol{X}_b, s).$$

To illustrate the performance of the permutation test, summarized in Algorithm 3, we simulated two datasets with $n = 400$ samples and $k = 10$ equally sized clusters. There are $s = 15$ informative featuures in each, while the first dataset has an ambient dimension of $p = 50$ while the second involves only $p = 20$ total features. These settings contrast sparse and dense feature-driven data; a detailed description of the simulation is described in Section 4. The true $s$ can be recovered in both settings, with gap statistics plotted in Figure 1.

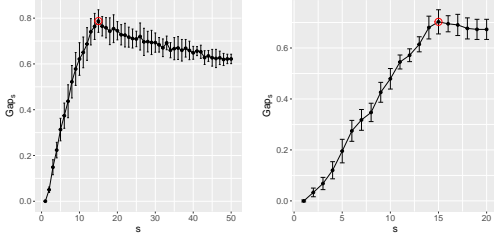

(a) Gap statistic when the number of noisy variables is much greater than the number of informative variables.

(b) Gap statistic when the number of noisy variables is much smaller than the number of informative variables.

Figure 1: Gap statistics in sparse and dense settings. The ground truth $s = 15$, labeled by the red circle in both plots, is selected by the permutation test in both settings.

---

**Algorithm 3** SKFR permutation tuning pseudocode

---

**Input:** data $\boldsymbol{X} \in \mathbf{R}^{n \times p}$, class size $k$, sparsity range $S$, number of permutation replicates $B$.
Create $B$ permuted datasets $\boldsymbol{X}_b$ by independently permuting the observations within each column of $\boldsymbol{X}$.
**for** each $s \in S$: **do**
    Compute the statistic $O(\boldsymbol{X}, s)$.
    **for** each $b = 1 : B$: **do**
        Perform SKFR with sparsity $s$ on $\boldsymbol{X}_b$.
        Compute $O(\boldsymbol{X}_b, s)$.
    **end for**
    Compute the gap statistic $\mathrm{Gap}(s) = \log O(\boldsymbol{X}, s) - \frac{1}{B} \sum_{b=1}^{B} \log O(\boldsymbol{X}_b, s)$.
**end for**
Choose $s \in S$ to maximize Gap(s).

---

## 4 Performance and simulation study

In this section, we use simulated experiments to validate SKFR and its extensions and to compare them to competing peer algorithms. We consider both locally and globally informative features to assess both variants, and then showcase SKFR's adaptability in handling missing data. In all simulations, the number of informative features $s$ is chosen to be 10, and we explore a range of sparsity levels by varying the total number of features $p \in (20, 50, 100, 200, 500, 1000)$. The SKFR variant and all the competing algorithms are seeded by the $k$-means++ initialization scheme [18]. We use the adjusted Rand index (ARI) [41, 42] to quantify the performance of the algorithms. These values are plotted for ease of visual comparison, while detailed numerical ARI results as well as analogous results in terms of normalized variation of information (NVI) [43] are tabulated in the Supplement. Varying the dimension $p$ from 20 to 1000, we run 30 trials with 20 restarts per trial. We tune SKM's $\ell_1$ bound parameter over the range $[2, 10]$ by the gap statistic.

**Sparse clustering test** In this experiment, we test SKFR1 against Lloyd's algorithm and SKM. The latter algorithm is implemented in the R package `sparcl`. We follow the simulation setup of Brodinová et al. [32], with details in the Supplement.

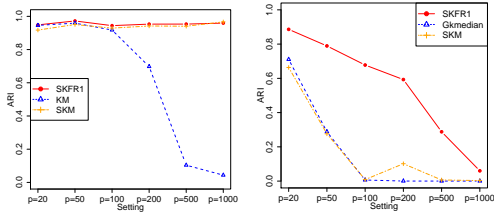

Figure 2: Median ARI with (left) and without (right) outliers.

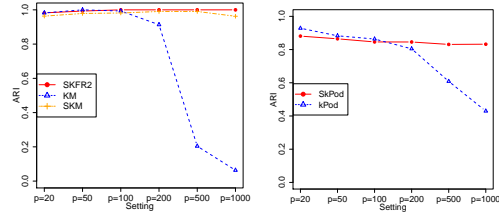

Figure 3: Subsets of discriminative features (left) and presence of missing data (right)

Figure 2 plots median ARI values over 30 trials under each simulation setting for the three algorithms. It is evident from the plot that Lloyd's $k$-means algorithm eventually deteriorates as the dimension of the dataset and number of noisy features increase. Both SKFR and SKM perform well without outliers, illustrating the utility of sparse clustering in high-dimensional settings.

In terms of feature selection, SKFR maintains false positive median rates very close to 0 over 30 trials under all 6 experimental settings. For SKM, the entries of the weight vector $w$ corresponding to the informative features have higher values than uninformative entries at convergence. However, SKM does not consistently deliver sparse solutions. For example, at convergence all entries of the weight vector $w$ are non-zero for the $p = 20$ setting, while for the $p = 1000$ setting over 400 entries are non-zero. Therefore, we find SKFR preferable to SKM in terms of feature selection.

Table 1: Runtime in seconds of SKFR1 and Lloyd's $k$-means, with (iteration count) below

|        | p= 20  | 50     | 100    | 200    | 500    | 1000   |
|--------|--------|--------|--------|--------|--------|--------|
| SKFR1  | **0.002** | **0.003** | **0.004** | **0.011** | **0.014** | 0.04   |
|        | **(6.0)** | (8.0)  | (10.5) | **(10.7)** | **(11.3)** | (17.0) |
| Lloyd  | 0.017  | 0.021  | 0.030  | 0.045  | 0.053  | **0.010** |
|        | (6.7)  | **(7.3)** | **(9.8)** | (11.8) | (13.3) | **(2.02)** |

Table 2: False Positive Rate of SKFR1 and SKM, with (false negative rate) below

|        | $p = 20$ | 50     | 100    | 200    | 500    | 1000   |
|--------|----------|--------|--------|--------|--------|--------|
| SKFR1  | **0.0**  | **0.025** | **0.028** | **0.016** | **0.012** | **0.009** |
|        | **(0.0)** | **(0.10)** | **(0.25)** | **(0.30)** | **(0.60)** | **(0.90)** |
| SKM    | 0.10     | 0.05   | 0.044  | 0.068  | 0.059  | 0.039  |
|        | (0.25)   | (0.60) | (1.0)  | (0.80) | (1.0)  | (1.0)  |

This experiment also allows us to examine runtime and iteration counts for SKFR and standard $k$-means. Table 1 tabulates our results obtained in a Julia 1.1 implementation [44]. For each simulation setting, the table shows average results over 20 runs. In almost all cases, SKFR requires much less run time. For $p = 1000$, $k$-means terminates first, but is unable to handle the high number of noisy features. In such cases, iterations until convergence can be misleading as $k$-means clearly stops short at a poor local minimum, as evident from Figure 2. Since methods such as SKM are of much higher computational complexity, as they call $k$-means as a *subroutine* per iteration, they are omitted from the timing comparison. Our empirical study emphasizes that SKFR stands out in terms of runtime—not only is it on par with $k$-means in terms of complexity, but in practice may converge faster.

**Robustness against outliers**  To test the robustness of SKFR to outliers, we adopt our earlier simulation settings and replace a modicum of the observations in each class by outliers, again following a design by Brodinová et al. [32] with details in the Supplement. For this experiment, we again generate $k = 10$ equally sized classes and a total of $n = 400$ observations. We compare SKFR1 to SKM, as well as the geometric $k$-median algorithm [45], which is designed to be robust to outliers compared to $k$-means. The latter algorithm is available in R in the Gmedian package. We generate 30 trials for each setting and use 20 restarts for each algorithm.

Median ARI values under outliers are plotted in the right panel of Figure 2. The geometric $k$-median algorithm deteriorates severely in the high dimensional cases, where there are many more noise features than informative features. To show SKFR's feature selection capability in the presence of outliers, we tabulate median false positive and false negative rates. Since SKM did not typically result in a sparse solution, we set a threshold and choose weight components greater than 0.1 to be selected features. In all outlier settings, SKFR1 is noticeably superior to SKM in feature selection accuracy.

**Clustering on different subsets of attributes**  We next consider simulated data with different sets of distinguishing attributes for different underlying classes. We generate $n = 250$ samples spread unevenly over $k = 5$ classes. The $s$ informative features of samples from each class are multivariate standard normal samples whose centers are initialized in the hypercube with entries $\theta_{jl} \sim 6 \cdot \mathcal{U}(0, 1)$ for $l = 1, \cdots, s$, and the $p - s$ uninformative features are sampled from $\mathcal{N}(0, 3)$. The set of informative features of each cluster is chosen by independently sampling a subset of size $s$ from all the $p$ dimensions. This simulation setting is modeled after a design that has appeared through numerous previous studies of $k$-means [46–48]. We compare the local version SKFR2 to Lloyd's $k$-means algorithm and SKM under the same levels of sparsity as the previous experiments. The median ARI results of the algorithms in this setting are plotted in the left panel of Figure 3.

Though SKM outperforms Lloyd's $k$-means significantly, we observe that our SKFR2 method performs at least on par with SKM, and begins to outperform when $p$ is large enough, despite its significantly lower computational cost.

**Partially observed data**   This experiment follows a similar data generation procedure with 10 informative features shared across all classes. $n = 250$ samples are generated and spread unevenly over $k = 5$ classes at random. The informative features are again multivariate standard normal samples with centers $\theta_{jl} \sim 6 \cdot \mathcal{U}(0, 1)$ for $l = 1, \cdots, s$, while the uninformative features are sampled from $\mathcal{N}(0, 1.5)$. To simulate missingness, $10\%$ of the entries of $\boldsymbol{X} = \{x_{ij}\}$ are replaced by `NA` values uniformly at random. We compare SKFR1 with imputation under the $k$-pod algorithm as implemented in the `kpodclustr` R package. For each setting we run 30 trials, and report the best of 10 random restarts for each algorithm. The median ARI results of the 30 trials for both algorithms are plotted in Figure 3, right panel. As in the complete data scenarios, SKFR remains competitive and actually outperforms $k$-pod for high-dimensional cases in the presence of noisy data.

## 5   Case study and real data benchmarks

**Mice protein data**   Having validated the algorithm on synthetic data, we begin our real data analysis by examining a mice protein expression dataset from a study of murine Down Syndrome [49]. The dataset consists of 1080 expression level measurements on 77 proteins measured in the cerebral cortex. There are 8 classes of mice (control versus trisomic, stimulated to learn versus not stimulated, injected with saline versus not injected). In the original paper, Higuera et al. [49] use self-organizing maps (SOMs) to discriminate subsets of the 77 proteins. The authors employ SOMs of size $7 \times 7$ to organize the control mice into 49 nodes. For classifying trisomic mice, the size of SOMs is chosen to be $6 \times 6$. In our treatment of the data, we use SKFR1 to cluster the control mice and the trisomic mice into 49 and 36 groups, respectively, and compare our clustering results with those obtained using Lloyd's algorithm, the SOMs from the original analysis, and the power $k$-means algorithm which was recently used to reanalyze this dataset [48]. We evaluate clustering quality following measures proposed by Higuera et al. [49]: number of mixed class nodes, which result from assigning mice from more than one class, and the total number of mice in mixed-class nodes. All methods are seeded with $k$-means++ over 20 initializations, and the sparsity level for SKFR1 is chosen to be $s = 24$ as tuned via the gap statistic for both the control and trisomic mice data. The classification results are displayed in Table 3, showing that SKFR typically achieves a noticeably higher clustering quality than its competitors.

In contrast to the other methods, our algorithm not only assigns labels to the mice, but also produces interpretable output informing us of the most discriminating proteins. Out of the 24 selected proteins for the control group, 18 proteins are consistent with those identified by Higuera et al. [49]; for the trisomic mice group, 20 proteins match those from the original study. The complete lists of informative proteins selected by SKFR and further details of our analysis appear in the Supplement.

Table 3: Protein expression level clustering quality from a mouse trisomy learning study; a lower number of mixed nodes indicates better performance [49].

|  | Control Mice | | Trisomic Mice | |
|---|---|---|---|---|
|  | Mixed Nodes | Total Mixed | Mixed Nodes | Total Mixed |
| SOM | 8 | 110 | 5 | 84 |
| SKFR1 | **4** | **67** | **3** | **64** |
| Power $k$-means | 7 | 92 | 4 | 70 |
| $k$-means++ | 11 | 164 | 9 | 152 |

**Benchmark datasets**   To further validate our proposed algorithm, we compare SKFR1 to widely used peer algorithms on 10 benchmark datasets collected from the Keel, ASU, and UCI machine learning repositories. A description of each dataset is given in the Supplement. Our algorithm competes with Lloyd's $k$-means algorithm, sparse $k$-means algorithm (SKM) from the R package `sparcl`, and entropy-weighted $k$-means [5] from the R package `wskm`. While `wskm` is not a sparse clustering method per se, it is a popular approach that assigns weights to feature relevance, and we do not know whether exact sparsity holds in these real data scenarios. On each example, we run 20 independent trials, in which each competing algorithm is given matched $k$-means++ initial seedings for fair comparison. Both SKM and SKFR1 are tuned using the gap statistic, and we run `wskm` using its default settings. We evaluate the performance of the algorithms on each dataset with normalized

mutual information (NMI) [43] and report the results in Table 4. The performance evaluated by ARI is given in the Supplement. SKFR1 gives the best result in terms of NMI for 9 of the 10 datasets.

Table 4: NMI values of SKFR1 and competing algorithms on data benchmarks, (standard deviation) below

|  | Newthyroid | WarpAR10P | WarpPIE10P | Iris | Wine | Zoo | WDBC | LIBRAS | Ecoli | Wall Robot 4 |
|---|---|---|---|---|---|---|---|---|---|---|
| SKFR1 | **0.441** | **0.189** | **0.251** | **0.815** | **0.729** | **0.825** | **0.585** | **0.565** | **0.367** | 0.176 |
|  | (0.059) | (0.056) | (0.031) | (0.032) | (0.054) | (0.047) | (0.002) | (0.024) | (0.041) | (0.065) |
| $k$-means++ | 0.348 | 0.187 | 0.240 | 0.732 | 0.414 | 0.703 | 0.421 | 0.564 | 0.363 | 0.155 |
|  | (0.101) | (0.027) | (0.045) | (0.048) | (0.018) | (0.066) | (0.001) | (0.022) | (0.042) | (0.021) |
| SKM | 0.214 | 0.131 | 0.229 | 0.797 | 0.416 | 0.705 | 0.421 | 0.551 | 0.362 | 0.155 |
|  | (0.001) | (0.025) | (0.054) | (0.003) | (0.016) | (0.054) | (0.001) | (0.017) | (0.014) | (0.088) |
| EW $k$-means | 0.205 | 0.183 | 0.190 | 0.667 | 0.406 | 0.683 | 0.384 | 0.486 | 0.366 | **0.246** |
|  | (0.066) | (0.020) | (0.042) | (0.156) | (0.029) | (0.072) | (0.080) | (0.030) | (0.039) | (0.092) |

## 6 Discussion

We present a novel framework (SKFR) for sparse $k$-means clustering with an emphasis on simplicity and scalability. In particular, SKFR does not rely on global shrinkage or $\ell_1$ penalization. Because its ranking phase is very quick, SKFR preserves the low time-complexity of Lloyd's algorithm. Further, it inherits properties such as monotonicity, finite-time convergence, and consistency. As noted previously, feature selection consistency is an interesting avenue for further theoretical investigation. In the context of optimization, if we treat the centroids as the optimization variables, then both the optimization variables and the set of nonsparse features are being greedily optimized in each iteration. In contrast, in existing optimization techniques such as projected gradient descent and iterative hard thresholding, the greedy optimization operation is applied to the optimization variable itself, emphasizing the novelty of our proposed framework.

We find empirically that SKFR is competitive with alternatives in sparse clustering despite incurring (often significantly) lower computational overhead. In contrast to its competitors, SKFR is highly scalable and modular. As demonstrated in Section 4, its extensions gracefully handle data settings including missing values and outliers. Finally, SKFR requires just one hyperparameter, the sparsity level $s$, which can be tuned using the gap statistic or cross-validation. In some cases, a desired or known level $s$ may be specified in advance. As parameter tuning in unsupervised settings is often costly and may suffer from instability, our approach can be quite advantageous. In contrast to applying generic dimension reduction before clustering, for instance pre-processing using principal component analysis (PCA), our approach also selects features that are interpretable in the original space—for instance, in identifying relevant genes—while PC space fails to preserve this interpretability.

The core idea of ranking and truncating relevant features can be viewed as a special case of clustering under a general set constraint $\boldsymbol{\theta}_j \in T$ on the cluster centers. Here the set $T \subset \mathbb{R}^p$ should be closed, but alternative constraints rather than the sparse ball may be relevant. Further exploration of this principle is warranted since it may be helpful in designing useful extensions and has proven successful for related estimation tasks [50–52]. In the cluster center recalculation step, for each cluster $C_j$ the loss can be improved by minimizing the criterion

$$\sum_{i \in C_j} \|\boldsymbol{x}_i - \boldsymbol{\theta}_j\|_2^2 = |C_j| \cdot \|\frac{1}{|C_j|} \sum_{i \in C_j} \boldsymbol{x}_i - \boldsymbol{\theta}_j\|_2^2 + d, \tag{6}$$

where $d$ is a constant irrelevant to the minimization process. The solution $\boldsymbol{\theta}_j = P_T \left( \frac{1}{|C_j|} \sum_{i \in C_j} \boldsymbol{x}_i \right)$ can be phrased in terms of the center of mass of the data points and the projection operator $P_T(\boldsymbol{y})$ taking a point $\boldsymbol{y}$ to the closest point in $T$. Many projection operators can be computed by explicit formulas or highly efficient algorithms [53–55], while the same principle may be employed for clustering under more general divergences such as Bregman divergences [56]. The notion of projection carries over and remains computationally tractable for geometrically simple yet relevant sets such as box constraints or halfspaces [9]. Thus, our novel framework and algorithmic primitive for sparse clustering apply to a broad class of constrained clustering problems. In our view, this is a fruitful avenue for future research.

## Broader Impact

Our work contributes theoretically and computationally to unsupervised learning and feature selection. Because we simply provide a more scalable and transparent algorithm for addressing a classical task,

our work brings no immediate ethical or societal concerns. Since our work focuses on improving clustering accuracy and preserving the interpretability and identifiability of features during and after the clustering process, it can help to reduce spurious conclusions drawn from off-the-shelf methods that either lack interpretability or are more likely to produce poor solutions. Indeed, as many scientific problems with societal impacts such as discovering gene associations for rare diseases or risk factors in precision medicine can be partially addressed through the lens of unsupervised learning, our work can positively impact these areas with wide implications. While our method does not leverage any bias in data in any way, misinterpreting or having too much confidence in solutions produced by our method on highly noisy, biased, or imbalanced data can potentially yield to inaccurate interpretations. We advise all users to understand the assumptions behind our method and the limitations of unsupervised learning, especially in challenging high-dimensional settings, when basing decisions off of the results of any machine learning approach such as that we propose here.

## Acknowledgments and Disclosure of Funding

This work was partially supported by NSF DMS 2030355, NHGRI HG006139 and NIH GM053275. We thank the anonymous referees and Saptarshi Chakraborty for their constructive comments and discussion.

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
