[Supplementary Material]

# Supplemental Material: Simple and Scalable Sparse *k*-means Clustering via Feature Ranking

**Zhiyue Zhang**[1]     **Kenneth Lange**[2]     **Jason Xu**[1]
[1]Department of Statistical Science, Duke University
[2] Departments of Computational Medicine, Statistics, and Human Genetics, UCLA

## 1  Proofs of Results in Section 3

We restate the results and provide their proofs below.

**Proposition 1.** *Each iteration of Alg. 1 and 2 monotonically decreases the objective* $h(C, \boldsymbol{\theta}) = \sum_{j=1}^{k} \sum_{\boldsymbol{x} \in C_j} \|\boldsymbol{x} - \boldsymbol{\theta}_j\|_2^2$.

We prove the descent property of Algorithm 1 here; the proof for Algorithm 2 follows analogously.

*Proof.* We will show that each step of the algorithm decreases the objective monotonically: that is, it suffices to show that $h(C^{(t+1)}, \boldsymbol{\theta}^{(t+1)}) \le h(C^{(t)}, \boldsymbol{\theta}^{(t)})$ where $t$ denotes iteration number. We will also use $h(C_j, \boldsymbol{\theta}_j)$ to refer to the contribution of only cluster $j$ to the $k$-means objective: that is, $h(C, \boldsymbol{\theta}) = \sum_{j=1}^{k} h(C_j, \boldsymbol{\theta}_j)$.

The label update satisfies a descent in the objective by definition: points are assigned to the nearest $\boldsymbol{\theta}_j$ (this step is identical to $k$-means), so that

$$h(C^{(t+1)}, \boldsymbol{\theta}^{(t)}) \le h(C^{(t)}, \boldsymbol{\theta}^{(t)}). \tag{1}$$

It remains to show the same for the centroid update:

$$h(C^{(t+1)}, \boldsymbol{\theta}^{(t+1)}) \le h(C^{(t+1)}, \boldsymbol{\theta}^{(t)}). \tag{2}$$

To this end, it will be useful to employ the following equality, which follows from the bias-variance decomposition:

**Lemma 1.** *For any assignment $C_j$ and centroid $\boldsymbol{\theta}_j$,*

$$h(C, \boldsymbol{\theta}) = h(C, \boldsymbol{\mu}_j) + |C_j| \|\boldsymbol{\theta}_j - \boldsymbol{\mu}_j\|_2^2, \qquad \textit{where } \boldsymbol{\mu}_j = \frac{1}{|C_j|} \sum_{\boldsymbol{x}_i \in C_j} \boldsymbol{x}_i.$$

We will also require some additional notation. Recall that between label and centroid updates, the algorithm chooses $s$ informative features by ranking each feature dimension $l$ according to the criterion

$$d_l = \sum_{j=1}^{k} |C_j| \mu_{jl}^2.$$

As the top $s$ features may change across iterations, let $S^{(t)}$ denote this set of top ranked "informative" dimensions at iteration $t$. Now we are ready to show that the objective function decreases under newly assigned sparse centers when labels are held fixed. We begin by expanding the left side of

Equation (2) according to Lemma 1:

$$h(C^{(t+1)}, \boldsymbol{\theta}^{(t+1)}) = \sum_{j=1}^{k} h(C_j^{(t+1)}, \boldsymbol{\mu}_j^{(t+1)}) + \sum_{j=1}^{k} |C_j^{(t+1)}| \|\boldsymbol{\theta}_j^{(t+1)} - \boldsymbol{\mu}_j^{(t+1)}\|_2^2$$

$$= b + \sum_{j=1}^{k} |C_j^{(t+1)}| \Big( \sum_{l \in S^{(t+1)}} \big(\theta_{jl}^{(t+1)} - \mu_{jl}^{(t+1)}\big)^2 + \sum_{l \notin S^{(t+1)}} \big(\theta_{jl}^{(t+1)} - \mu_{jl}^{(t+1)}\big)^2 \Big)$$

$$= b + \sum_{j=1}^{k} |C_j^{(t+1)}| \Big( 0 + \sum_{l \notin S^{(t+1)}} \big(\mu_{jl}^{(t+1)} - 0\big)^2 \Big)$$

$$= b + \sum_{l \notin S^{(t+1)}} \sum_{j=1}^{k} |C_j^{(t+1)}| \big(\mu_{jl}^{(t+1)}\big)^2$$

$$= b + \sum_{l \notin S^{(t+1)}} d_l^{(t+1)},$$

where $b = \sum_{j=1}^{k} h(C_j^{(t+1)}, \boldsymbol{\mu}_j^{(t+1)})$ is a constant with respect to $\boldsymbol{\theta}$. The third equality holds because note the update rule at step $t+1$ sets each $\boldsymbol{\theta}_j = \boldsymbol{\mu}_j$ along the top $s$ features, while the sparsity projection replaces the components along uninformative features $l \notin S^{(t+1)}$ by zero.

Next, we consider a similar treatment of the right hand side of Equation (2): prior to the centroid update,

$$h(C^{(t+1)}, \boldsymbol{\theta}^{(t)}) = \sum_{j=1}^{k} h(C_j^{(t+1)}, \boldsymbol{\mu}_j^{(t+1)}) + \sum_{j=1}^{k} |C_j^{(t+1)}| \|\boldsymbol{\theta}_j^{(t)} - \boldsymbol{\mu}_j^{(t+1)}\|_2^2$$

$$= b + \sum_{j=1}^{k} |C_j^{(t+1)}| \sum_{l \in S^{(t)}} \big(\theta_{jl}^{(t)} - \mu_{jl}^{(t+1)}\big)^2 + \sum_{l \notin S^{(t)}} \sum_{j=1}^{k} |C_j^{(t+1)}| \big(\mu_{jl}^{(t+1)} - 0\big)^2$$

$$= b + \sum_{j=1}^{k} |C_j^{(t+1)}| \sum_{l \in S^{(t)}} \big(\mu_{jl}^{(t)} - \mu_{jl}^{(t+1)}\big)^2 + \sum_{l \notin S^{(t)}} d_l^{(t+1)}$$

$$\geq b + \sum_{l \notin S^{(t)}} d_l^{(t+1)} \geq b + \sum_{l \notin S^{(t+1)}} d_l^{(t+1)} = h(C^{(t+1)}, \boldsymbol{\theta}^{(t+1)}).$$

The final inequality follows by noting that $\sum\limits_{l \notin S^{(t+1)}} d_l^{(t+1)}$ sums over the $p - s$ lowest ranked features, so it must bound any sum over $p - s$ features $\sum\limits_{l \notin S^{(t)}} d_l^{(t+1)}$ from below.

Hence we arrive at Equation (2). Together with Equation (1), we thus conclude that the objective function $h(C, \boldsymbol{\theta})$ monotonically decreases at each iteration. $\square$

**Proposition 2.** *Assume that for any neighborhood $\mathcal{N}$ of $\Theta^*$, there exists $\eta > 0$ such that $\Phi(\Theta, P) > \Phi(\Theta^*, P) + \eta$ for every $\Theta \notin \mathcal{N}$. Then if $\Theta_n$ eventually lie in the same dimension as $\Theta^*$, we have $\Theta_n \xrightarrow{a.s.} \Theta^*$ as $n \to \infty$.*

*Proof.* The proof builds on the original $k-$means consistency proof by Pollard [1], in which the author showed the uniform SLLN: Let $\mathcal{G}$ denote the family of $P-$integrable functions of the form

$$g_\Theta(\boldsymbol{x}) = \min_{\boldsymbol{\theta} \in \Theta} \|\boldsymbol{x} - \boldsymbol{\theta}\|_2^2;$$

then

$$\sup_{g \in \mathcal{G}} \Big| \int g \, dP_n - \int g \, dP \Big| \to 0 \quad \text{a.s.} \tag{3}$$

Because our algorithm shares an objective with $k$-means, we may appeal to Equation (3) directly. To this end, we want to show that given any neighborhood $\mathcal{N}$ of $\Theta^*$, there exists $M > 0$ such that $\Theta_n \in \mathcal{N}$ almost surely whenever $n > M$. Because we have assumed selection consistency so that the dimension of $\Theta_n$ is eventually fixed to that of $\Theta^*$, it is enough to show that for any $\eta > 0$, there exists $M > 0$ such that $\Phi(\Theta_n, P) \leq \Phi(\Theta^*, P) + \eta$ for all $n > M$.

Observe that we may divide the relevant term in to three parts

$$\Phi(\Theta_n, P) - \Phi(\Theta^*, P) = \xi_1 + \xi_2 + \xi_3,$$

where

$$\xi_1 = \Phi(\Theta_n, P_n) - \Phi(\Theta^*, P_n), \quad \xi_2 = \Phi(\Theta_n, P) - \Phi(\Theta_n, P_n), \quad \xi_3 = \Phi(\Theta^*, P_n) - \Phi(\Theta^*, P).$$

Note that $\xi_1 \leq 0$ is bounded above by zero from its definition. On the other hand, both of the remaining $\xi_2$ and $\xi_3$ can be bounded by $\eta/2$ by appealing to uniform strong law Equation (3) established in [1]. The result follows. $\qquad\square$

## 2   Description of the first two numerical experiments in Section 5

**Sparse clustering test**   For this experiment, we follow the simulation setup of Brodinová et al. [2]. We have $n = 400$ samples spread over $k = 10$ classes of equal size. The informative features of samples from class $j$ follow a multivariate normal distribution $\mathcal{N}(\boldsymbol{\mu}_j, \boldsymbol{\Sigma}_j)$. The mean vector $\boldsymbol{\mu}_j$ is constructed by first sampling a scalar $\mu$ from the distribution $\mathcal{U}[-6, -3] \cup \mathcal{U}[3, 6]$. If we assume for convenience that the informative features are numbered $1, \ldots, s$, then we set

$$\mu_{jl} = \begin{cases} \mu, & l \in \{a_{j1}, a_{j2}, \ldots, \} \\ 0, & \text{otherwise}, \end{cases} \tag{4}$$

where $a_{jn}$ is the sequence defined recursively by $a_{j1} = j$ and $a_{j(n+1)} = a_{jn} + k$. For instance, if we consider $k = 3$ groups of 7 dimensions, then we would have $\boldsymbol{\mu}_1 = (\mu, 0, 0, \mu, 0, 0, \mu)$, $\boldsymbol{\mu}_2 = (0, \mu, 0, 0, \mu, 0, 0)$, and $\boldsymbol{\mu}_3 = (0, 0, \mu, 0, 0, \mu, 0)$. The covariance matrix $\boldsymbol{\Sigma}_j$ is generated [3] as

$$\boldsymbol{\Sigma}_j = \boldsymbol{Q} \begin{pmatrix} 1 & \rho_j & \cdots & \rho_j \\ \rho_j & \ddots & \ddots & \vdots \\ \vdots & \ddots & \ddots & \rho_j \\ \rho_j & \cdots & \rho_j & 1 \end{pmatrix} \boldsymbol{Q}^\top, \tag{5}$$

where $\boldsymbol{Q}$ is a random rotation matrix with $\boldsymbol{Q}^\top = \boldsymbol{Q}^{-1}$, and the value $\rho_j$ in the off-diagonal entries is chosen uniformly over the interval $[0, 1, 0.9]$. The $p - s$ uninformative features are sampled from univariate standard normal distributions.

**Robustness against outliers**   we adopt our earlier simulation settings and replace a modicum of the observations in each class by outliers. Specifically, we contaminate 2 informative features in 10% of the observations with scatter outliers generated according to the $\mathcal{N}(\boldsymbol{\mu}_j, \sigma \boldsymbol{I})$ distribution, where $\sigma \sim \mathcal{U}[3, 10]$, and $\boldsymbol{\mu}_j$ is the mean vector described in Equation (4). Next, we contaminate 10% of the uninformative features in another 10% of the observations by uniformly distributed outliers $\sim \mathcal{U}[-12, 6] \cup \mathcal{U}[6, 12]$. For this experiment, we again generate $k = 10$ equally sized classes and a total of $n = 400$ observations.

## 3   Detailed tables for results and performance in Section 5

Table 1: Adjusted Rand Index (ARI) Best of result, and (median) below for Section 5.1

|  | $p = 20$ | $p = 50$ | $p = 100$ | $p = 200$ | $p = 500$ | $p = 1000$ |
|---|---|---|---|---|---|---|
| SKFR | 1.000 (**0.949**) | 1.000 (**0.972**) | 1.000 (**0.944**) | 0.994 (**0.953**) | 1.000 (**0.953**) | 1.000 (0.959) |
| Lloyd | 1.000 (0.946) | 1.000 (0.959) | 0.994 (0.918) | 0.983 (0.700) | 0.703 (0.104) | 0.114 (0.044) |
| SKM | 1.000 (0.917) | 1.000 (0.951) | 0.994 (0.929) | 0.994 (0.942) | 1.000 (0.940) | 1.000 (**0.967**) |

Table 2: Normalized Variation of Information (NVI) Best of result, and (median) below for Section 5.1

|  | $p = 20$ | $p = 50$ | $p = 100$ | $p = 200$ | $p = 500$ | $p = 1000$ |
|---|---|---|---|---|---|---|
| SKFR | 0 (**0.077**) | 0 (**0.047**) | 0 (**0.079**) | 0.010 (**0.067**) | 0 (**0.076**) | 0 (0.059) |
| Lloyd | 0 (0.082) | 0 (0.064) | 0.010 (0.115) | 0.027 (0.332) | 0.335 (0.885) | 0.881 (0.934) |
| SKM | 0 (0.118) | 0 (0.073) | 0.010 (0.104) | 0.010 (0.087) | 0 (0.091) | 0 (**0.054**) |

Table 3: Adjusted Rand Index (ARI) Best of result, and (median) below for Section 5.2

|  | $p = 20$ | $p = 50$ | $p = 100$ | $p = 200$ | $p = 500$ | $p = 1000$ |
|---|---|---|---|---|---|---|
| SKFR | 0.994 (**0.886**) | 0.972 (**0.789**) | 0.989 (**0.678**) | 0.972 (**0.593**) | 0.978 (**0.288**) | 1.000 (**0.060**) |
| Gmedian | 0.921 (0.711) | 0.710 (0.288) | 0.124 (0.006) | 0.0005 (0.0002) | 0.0002 (0.0001) | 0.0002 (0.0001) |
| SKM | 0.989 (0.664) | 0.875 (0.276) | 0.504 (0.009) | 0.493 (0.101) | 0.359 (0.007) | 0.098 (0.003) |

Table 4: Normalized Variation of Information (NVI) Best of result, and (median) below for Section 5.2

|  | $p = 20$ | $p = 50$ | $p = 100$ | $p = 200$ | $p = 500$ | $p = 1000$ |
|---|---|---|---|---|---|---|
| SKFR | 0.010 (**0.115**) | 0.047 (**0.243**) | 0.020 (**0.332**) | 0.044 (**0.426**) | 0.040 (**0.669**) | 0 (**0.905**) |
| Gmedian | 0.118 (0.325) | 0.302 (0.649) | 0.753 (0.941) | 0.966 (0.972) | 0.976 (0.978) | 0.976 (0.978) |
| SKM | 0.020 (0.378) | 0.122 (0.720) | 0.941 (0.967) | 0.431 (0.858) | 0.543 (0.964) | 0.820 (0.966) |

Table 5: Adjusted Rand Index (ARI) Best of result, and (median) below for Section 5.3

|  | $p = 20$ | $p = 50$ | $p = 100$ | $p = 200$ | $p = 500$ | $p = 1000$ |
|---|---|---|---|---|---|---|
| SKFR | 1.000 **(0.982)** | 1.000 (0.993) | 1.000 (**1.000**) | 1.000 (**1.000**) | 1.000 (**1.000**) | 1.000 (**1.000**) |
| Lloyd | 1.000 (0.982) | 1.000 (**1.000**) | 1.000 (0.993) | 1.000 (0.913) | 0.734 (0.204) | 0.163 (0.063) |
| SKM | 1.000 (0.963) | 1.000 (0.979) | 1.000 (0.982) | 1.000 (0.990) | 1.000 (0.990) | 1.000 (0.963) |

Table 6: Normalized Variation of Information (NVI) Best of result, and (median) below for Section 5.3

|  | $p = 20$ | $p = 50$ | $p = 100$ | $p = 200$ | $p = 500$ | $p = 1000$ |
|---|---|---|---|---|---|---|
| SKFR | 0 ( **0.044)** | 0 (0.023) | 0 (**0**) | 0 (**0**) | 0 (**0**) | 0 (**0**) |
| Lloyd | 0 (0.048) | 0 (**0**) | 0 (0.023) | 0 (0.181) | 0.433 (0.865) | 0.888 (0.950) |
| SKM | 0 (0.085) | 0 (0.055) | 0 (0.043) | 0 (0.025) | 0 (0.024) | 0 (0.094) |

Table 7: Adjusted Rand Index (ARI) Best of result, and (median) below for Section 5.4

|  | $p = 20$ | $p = 50$ | $p = 100$ | $p = 200$ | $p = 500$ | $p = 1000$ |
|---|---|---|---|---|---|---|
| SKFR | 0.963 (0.881) | 0.956 (0.865) | 0.954 (0.846) | 0.916 (**0.846**) | 0.941 (**0.831**) | 0.932 (**0.832**) |
| $k$-pod | 0.990 (**0.928**) | 0.963 (**0.883**) | 0.947 (**0.863**) | 0.883 (0.804) | 0.796 (0.609) | 0.746 (0.429) |

Table 8: Normalized Variation of Information (NVI) Best of result, and (median) below for Section 5.4

|  | $p = 20$ | $p = 50$ | $p = 100$ | $p = 200$ | $p = 500$ | $p = 1000$ |
|---|---|---|---|---|---|---|
| SKFR | 0.088 (0.238) | 0.116 (0.263) | 0.115 (0.277) | 0.183 (**0.290**) | 0.131 (**0.296**) | 0.138 (**0.288**) |
| $k$-pod | 0.025 (**0.144**) | 0.094 (**0.214**) | 0.117 (**0.264**) | 0.233 (0.340) | 0.344 (0.535) | 0.450 (0.692) |

# 4 Details of the real data studies in Section 6

The mouse protein dataset studied in Higuera et al. [4] is comprised of expression level measurements across 77 proteins that produce detectable signals in the nuclear fraction of the cortex. These measurements are obtained on 34 trisomic mice—those with Ts65Dn Down syndrome— and 38 wild-type (control group) mice, for a total of 72 mice. The dataset consists of $1080$ total measurements, with $510$ observations among the trisomic mice and $570$ pertaining to the controls.

The data pre-processing steps exactly follow that in [4], discarding one mouse with irregular measurements and using the same normalization across columns so that all measurements lie in the interval $[0, 1]$. When we assess partially observed data using our imputation algorithm, the missing data do not need to be in-filled via arbitrary rule. Otherwise, missing measurements in [4] are replaced with the average expression level of the corresponding protein within the same class of mice.

We consider self-organizing maps (SOM) because this was the unsupervised tool used in the original study. Higuera et al. choose a $7 \times 7$ self-organizing map to cluster control mice, and a $6 \times 6$ SOM to cluster the trisomic mice [4]. Note that the computational complexity for SOMs is greater by a factor of $n$ compared to our method, yet our method outperforms by the same metric. The number of nodes in the SOM plays the analogous role to the number of centers $k$.

The main text discusses the feature selection results and consistency with previous studies. Here we detail the selected proteins for each mouse for completeness.

For the control mice, the selected proteins are DYRK1A (c1,c2,c3,c4), ITSN1 (c1,c2,c3,c4), pAKT (FP), pCAMKII (c2,c4,c5), pERK (c1,c2,c3,c4), pJNK (FP), PKCA (c1,c4,c6), pMEK (FP), pP-KCAB (c1,c4,c6), BRAF (c1,c2,c3,c4), CAMKII (FP), GSK3B (c1,c2,c3,c4), SOD1 (c1,c2,c3,c4,c5), pNUMB (c1,c2,c3,c4,c5), pGSK3B (c1,c2,c3,c4), pPKCG (c5), CDK5 (c1,c2,c3,c4), S6 (c1,c2,c3,c4), AcetylH3K9 (FP), Tau (FP), PSD95 (c1,c4), Ubiquitin (c1,c2,c4,c5,c6), pGSK3B_Tyr216 (c3), CaNA (c1,c2,c3,c4). c1 through c6 correspond to the six columns of the supplemental table S1 of [4], showing discriminant proteins for each pair of different classes of control mice. For example, c1 corresponds to c-CS-s vs. c-SC-s, that is, control mice, stimulated to learn, injected with saline, vs. control mice, not stimulated to learn, injected with saline. For more details, please refer to the Supplement of [4]. FP denotes "false positive", meaning that those proteins that we found were not present in the table S1. It is worth noting that we were also able to identify proteins such as pERK and BRAF, which are well-known to be critical to learning [5].

For the trisomic mice, the selected proteins are DYRK1A (c1,c2,c3,c5), ITSN1 (c1,c2,c3), pCAMKII (FP), pERK (c1,c2,c3,c5), pJNK (c2), PKCA (FP), pMEK (c2,c3), pPKCAB (c4), pRSK (FP), BRAF (c1,c2,c3,c5), MTOR (c1,c2,c3), P38 (c1,c2,c3), pMTOR (c2,c4), DSCR1 (c2), NR2B (c1,c2), pNUMB (FP), RAPTOR (c2), pP70S6 (c1,c3,c4), pPKCG (c4), AcetylH3K9 (c3), TAU (c3), Ubiquitin (c2,c3,c5), H3AcK18 (c2,c3), CaNA (c2,c3). Similarly, c1 through c5 represent the columns of the supplemental table S2 of [4], which is a table for discriminant proteins for each pair of different classes of trisomic mice. We observe that BRAF, DYRK1A, ITSN1, P38, pERK and SOD1 all get selected; these 6 proteins are significantly in t-CS-s vs. t-SC-s, and all of them also differ in the corresponding set of control mice in normal learning, c-CS-s vs. c-SC-s.

## 4.1 Source and description of the benchmark datasets

Table 9: Source and description of the benchmark datasets

| Dataset | Source | k | n | p |
|---|---|---|---|---|
| Newthyroid | Keel Repository | 3 | 215 | 5 |
| WarpAR10P | ASU Repository | 10 | 130 | 2400 |
| WarpPIE10P | ASU Repository | 10 | 210 | 2420 |
| Iris | Keel Repository | 3 | 150 | 4 |
| Wine | Keel Repository | 3 | 178 | 13 |
| Zoo | Keel Repository | 7 | 101 | 16 |
| WDBC | UCI Repository | 2 | 569 | 30 |
| Movement Libras | Keel Repository | 15 | 360 | 90 |
| Ecoli | Keel Repository | 8 | 336 | 7 |
| Wall Robot 4 | UCI Repository | 4 | 5456 | 4 |

## 4.2 Benchmark datasets performance evaluated by ARI

Table 10: ARI values of SKFR and competing algorithms on benchmark datasets, and (standard deviation) below. We see the same trends as indicated by NMI in the main text.

|  | SKFR | k-means++ | SKM | EW k-means |
|---|---|---|---|---|
| Newthyroid | **0.573** | 0.462 | 0.254 | 0.133 |
|  | (0.019) | (0.170) | (0.029) | (0.086) |
| WarpAR10P | **0.033** | 0.026 | 0.003 | 0.025 |
|  | (0.040) | (0.034) | (0.021) | (0.023) |
| WarpPIE10P | **0.079** | 0.065 | 0.056 | 0.051 |
|  | (0.025) | (0.020) | (0.018) | (0.012) |
| Iris | **0.840** | 0.661 | 0.768 | 0.619 |
|  | (0.048) | (0.120) | (0.022) | (0.189) |
| Wine | **0.729** | 0.362 | 0.364 | 0.355 |
|  | (0.059) | (0.011) | (0.008) | (0.026) |
| Zoo | **0.771** | 0.644 | 0.643 | 0.575 |
|  | (0.085) | (0.089) | (0.055) | (0.111) |
| WDBC | **0.657** | 0.490 | 0.490 | 0.445 |
|  | (0.007) | (0.001) | (0.001) | (0.158) |
| LIBRAS | **0.311** | 0.301 | 0.284 | 0.223 |
|  | (0.019) | (0.018) | (0.015) | (0.030) |
| Ecoli | **0.344** | 0.320 | 0.285 | 0.337 |
|  | (0.064) | (0.060) | (0.019) | (0.068) |
| Wall Robot 4 | 0.091 | 0.076 | 0.074 | **0.141** |
|  | (0.099) | (0.008) | (0.010) | (0.084) |