[Reviews · NeurIPS 2020]

Review 1

Summary and Contributions: The paper is about unseupervised learning and the authors proposed sparse k-means clustering called SKRF. The authors claimed that SKRF is scalable and easy to implement. SKRF is evaluated on both synthetic and real data sets. However, my qesttiion is why authors didn't compare their method with stronger baselines? The selection of baselines needs revision. For instance, why authors didn't compare SKRF with reference [37] which has been published in ICML recently? +++++++++++++++++++++ After reading the author's response, I vote for acceptance since the authors addressed my misunderstanding.

Strengths: The paper is well-written and easy to understand. The topic is relevant to NeurIPS communitty but the contribution of the paper is limited. Authors need to compare their method with other strong baselines. Also they need to mentioned what is the benefit of SKRF over low-rank approximation like matrix or tensor factorization techniques.

Weaknesses: The contribution of this paper is very incremental and not enough for Neurips conference. Authors could pick better baselines for comparison. For instance, why they didn't compare their method with [37] "Power k-Means Clustering' which is recently published in ICML. Moreover, it is not clear the benefit of proposed technique over matrix and tensor factorization approaches?

Correctness: Minor issue: Capttion of Figure 3: feattures -> features

Clarity: The paper is well-written but the contribution is limited and not enough for Neurips Conference.

Relation to Prior Work: No, the difference with previiou works is not clear. Authors need to compare their proposed method with recent baselines.

Reproducibility: Yes

Additional Feedback:


Review 2

Summary and Contributions: This paper focuses on the problem of clustering in high dimension. K-means clustering is an extremely popular tool (especially in biomedical applications). However, as underlined by the authors, its performance is severely hindered in high-dimensional space --- leaving the data analyst no chance but to (a) apply some dimensionality reduction technique before performing the clustering or (b) selecting the features that are the most informative for the clustering and apply k-means on a subset of the features. This paper proposes a version of the later approach, choosing a sparse and interpretable subset of features. The setting is the following. Consider a dataset of d-dimensional observations (variables have been standardized.). The authors compute a score for each variable, which is basically the total variance of the observed values for that variable within each cluster. The $s$ variables with lower scores are used to assign variables to clusters (standard Lloyd’s step). In a way, this score is indicative of which variable achieve better clustering (less spread out cluster) as a measure of the relevance of the observations. The authors then prove the consistency of their added step with standard clustering, the convergence of their algorithm to the optimum (under the constraint that the features have been properly selected) and compare their method to other feature-selection methods for clustering .

Strengths: 1) Extremely simple and scalable algorithm: the solution proposed by the authors is simple, scalable, and intuitive. Its simplicity also allows it to benefit from theoretical guarantees for kmeans. 2) Ease of use: because $s$ is specified, we are ensured to select $s$ number of variables -- as opposed to using l1 penalty, where the effect of the penalty on the number of features selected is less obvious 3) Interpretable: since we are selecting features, (and not linear combination of features or lower embeddings, etc), the features that are selected are interpretable. This is great for biological applications, where we really wish to understand which features cause the clustering. 4) Robustness: Problems that arise in real data are tackled: the authors show the ease of adaptation of their method to the issues that plague real data analysis (missing data, outliers) 5) Experiments are well designed: the experiments that are led in this paper are very convincing --- both testing the approach on simulated data and on real data, reporting the standard deviation.

Weaknesses: Very convincing, theoretically sound, and potentially very useful paper, I found no real weakness. Maybe, it would have been interesting to discuss the use of their score (a sort of within-cluster sum of squares) compared to other metrics for computing the relevance of the features (like the entropy, mentioned in prior work), etc.

Correctness: Yes.

Clarity: Very well (see comment above)

Relation to Prior Work: Very clearly.

Reproducibility: Yes

Additional Feedback:


Review 3

Summary and Contributions: In this work, the authors describe a simple algorithm for sparse k-means clustering using feature ranking. The method is simple, generic and seems to provide a nice addition to the set of k-means related algorithms.

Strengths: The manuscript is well written and presents a simple but generic procedure to combine feature selection with K-means clustering. The authors showcase the strengths of their method on a number of applications.

Weaknesses: The major weak part of the paper is the experimental part. There, some more explanations about the the different datasets used would be welcome, as well as comparisons to other unsupervised feature selection methods.

Correctness: Claims are mostly correct, as well as the methodology.

Clarity: Very well written paper, easy to read.

Relation to Prior Work: While the authors discuss other related work, I am missing some comparisons to other unsupervised feature selection techniques (e.g. COSA). It would also be nice if some comparisons to purely filter-based unsupervised feature selection methods could be compared to.

Reproducibility: Yes

Additional Feedback: The authors claim the sorting step can be done in O(n) time, but this should be O(n log n)


Review 4

Summary and Contributions: This paper modifies the k-means objective by sorting features in a rank order such that the distance from 0 is maximized while the cluster membership distance is minimized. By doing so, the algorithm has a controllable sparsity. Two variants are presented, one that selects features based on all clusters and another that selects features per cluster.

Strengths: I enjoyed this paper. There are proofs in the supplement and the empirical evaluation is there. The discussion of extending the method to other projection operators is also interesting. This paper seems relevant to the community.

Weaknesses: One possible weakness is the claim that the selected features are more interpretable (282). Perhaps a table comparing some features selected by L1 vs ranking could be shown as an example to the reader how they are more interpretable vs domain specific examples like proteins. Maybe from a text corpus where k-means has traditionally been terrible because of the sparse high dimensionality of the features. The other use of the word interpretable when referring to the sparsity structure is also overloaded. Perhaps controllable sparsity structure is a better term. I have no idea what interpretable sparsity structure even means and it conflates the two uses of the word.

Correctness: The metric of NMI and the data sets used seem reasonable to me.

Clarity: Yes, I enjoyed this paper. It was written clearly and can be understood by many attendees of the conference.

Relation to Prior Work: Yes, the authors compare previous work and specifically compare to an L1 lasso version of the algorithm vs their ranking method.

Reproducibility: Yes

Additional Feedback: Post rebuttal I still like the paper and stick to my score

[Author Response · NeurIPS 2020]

Reviewers remark our method is intuitive and correct, and opens new directions in sparse clustering, while R1 raised
a concern about the extent of our contributions. Most comments mention the paper is well written while providing
constructive suggestions to further improve the presentation. We thank the referees for their time and feedback, and
provide detailed responses below:

R1:
We thank you for commenting the paper is well-written and for finding a typo. We hope you might reconsider the
novelty of our work if you find these responses to sufficiently address your concerns:
− You suggest better baselines for comparison, citing Power $k$-means [37] and matrix + tensor factorization. Our
manuscript *does* cite [37], and does compare with this method—see Table 3. Because power $k$-means is not designed
for the high-dimensional setting, it performed nearly identically with $k$-means in the simulated experiments where we
know the ground truth is sparse, and was omitted due to redundancy. We are happy to include its performance in those
comparisons as well, which will convey the same trends.
− Your intuition for low-rank matrix factorization is spot on, but low-rank factorization plus the constraint that one
seeks "hard" label assignments becomes **equivalent** to $k$-means, which we consider here. See for instance "$k$-means
Clustering Is Matrix Factorization" (Bauckhage 2015). Note that the number of clusters $k$ is analogous to the rank
$k$ of the low-rank factor; it remains nontrivial to perform feature selection jointly (i.e. simultaneously seek a sparse
number $s$ of informative features) as we do in the proposed method. In light of your comment, we will emphasize the
connection to matrix factorization in the revision. We do not compare to tensor factorization as all data we consider are
vector-valued and not matrix-valued.
− We respectfully disagree that the contribution is incremental, as the ranking-based feature selection is a marked
departure from the existing efforts which largely either rely on generic dimension reduction as a pre-processing step, or
penalization via norm-shrinkage. Instead, the proposed method allows for the exact desired number of nonzero features
to be specified as input, and yields a scalable approach that is appropriate in high-dimensional settings yet comparable
to Lloyd's algorithm in terms of simplicity and speed. As a result, our algorithm looks and functions quite differently
than past work on sparse $k$-means, which we have reviewed and compared to our method.

R2:
We thank you for your detailed comments and careful reading of the paper. We will further elaborate on the choice of
within-cluster sum of squares score as suggested during the revision.

R3:
We agree that the benchmarks and simulation details can be better described and will provide complete details in
the revision. You raise a good point about further competing methods– regarding COSA, we were unable to find the
authors' implementation and had implemented our own version whose runtime and performance was far worse than the
proposed method. We should also note that COSA was designed for feature weighing rather than selection, and does
not typically result in sparse solutions, though we will attempt to add a fair, detailed comparison in the revision. We
also note your suggestion regarding filter methods that focus only on feature selection and will include a comparison in
the final version.

R4:
You are absolutely correct that "interpretable sparsity" is overloaded here. We will clarify in the revision its twofold
meaning: first as you've mentioned, we can inform or "control" the sparsity level via parameter $s$. In this sense the
parameter $s$ is directly interpretable compared to parameters such as $\lambda$ in existing $\ell_1$ approaches. Second, our ranking
method selects features among the original dimensions, thus allowing them to retain their original interpretation. For
instance, in our mouse protein study it is important that the top ranked features identified by SKFR identify the most
relevant genes, as the original features correspond to expression levels along a high-dimensional space of candidate
genes. In this sense the *dimension reduction* is interpretable, in contrast to generic dimension reduction such as PCA,
where the axes (principal components) in the projected space lose their interpretation as genes. We will improve the
exposition to emphasize this, as well as provide further detail in a comparison table with sparse $k$-means focusing on
selection in the Supplement of the final draft, and thank you for these constructive comments.

[Meta-Review · NeurIPS 2020]

The reviewers appreciate the algorithmic contributions of this paper and believe it will be on interest to the community.